# TRH Regulates the Synthesis and Secretion of Prolactin in Rats with Adenohypophysis through the Differential Expression of miR-126a-5p

**DOI:** 10.3390/ijms232415914

**Published:** 2022-12-14

**Authors:** Guo-Kun Zhao, Yi Zheng, Hai-Xiang Guo, Hao-Qi Wang, Zhong-Hao Ji, Tian Wang, Song Yu, Jia-Bao Zhang, Bao Yuan, Wen-Zhi Ren

**Affiliations:** Department of Laboratory Animals, College of Animal Sciences, Jilin University, Changchun 130062, China

**Keywords:** prolactin, thyrotropin-releasing hormone, GH3 cell line, microRNA, pituitary gland, animal reproduction

## Abstract

Prolactin (PRL) is an important hormone that is secreted by the pituitary gland and plays an important role in the growth, development and reproduction of organisms. Thyrotropin-releasing hormone (TRH) is a common prolactin-releasing factor that regulates the synthesis and secretion of prolactin. In recent studies, microRNAs (miRNAs) have been found to play a key role in the regulation of pituitary hormones. However, there is a lack of systematic studies on the regulatory role that TRH plays on the pituitary transcriptome, and the role of miRNAs in the regulation of PRL synthesis and secretion by TRH lacks experimental evidence. In this study, we first investigated the changes in PRL synthesis and secretion in the rat pituitary gland after TRH administration. The results of transcriptomic analysis after TRH treatment showed that 102 genes, including those that encode Nppc, Fgf1, PRL, Cd63, Npw, and Il23a, were upregulated, and 488 genes, including those that encode Lats1, Cacna2d1, Top2a, and Tfap2a, were downregulated. These genes are all involved in the regulation of prolactin expression. The gene expression of miR-126a-5p, which regulates the level of PRL in the pituitary gland, was screened by analysis prediction software and by a dual luciferase reporter system. The data presented in this study demonstrate that TRH can regulate prolactin synthesis and secretion through miR-126a-5p, thereby improving our understanding of the molecular mechanism of TRH-mediated PRL secretion and providing a theoretical basis for the role of miRNAs in regulating the secretion of pituitary hormones.

## 1. Introduction

Anterior pituitary cells produce and secrete the peptide hormone prolactin (PRL), which is crucial for sustaining milk output, boosting milk synthesis, and controlling mammary gland growth during animal reproduction [1]. In addition to lactation, prolactin has now been reported to have more than 300 different functions in various vertebrates [2]. During animal reproduction, prolactin promotes luteogenesis while maintaining luteinizing secretion and the function of the corpus luteum [3]. Prolactin has a function in reproductive processes in addition to its roles in immune system control, maintenance of homeostatic osmotic pressure, and maintenance of internal environment stability [4,5]. The expression of prolactin is regulated by several factors, including estradiol, TRH, epidermal EGF, and dopamine. It is known that the hypothalamus is the primary regulator of pituitary hormone secretion [6].

The hypothalamus regulates the synthesis and secretion of pituitary prolactin through the secretion of prolactin inhibitory factors (PIFs) and prolactin-releasing factors (PRFs) [7]. Dopamine is a common inhibitor of prolactin secretion [8], and TRH is a neuropeptide secreted by the hypothalamus [9]. In mammals, TRH acts as a PRF that promotes prolactin release [10], and it is regulated by a cascade of several different signaling pathways, including those that regulate TRH-induced prolactin synthesis and secretion. When TRH is coupled to the Gq protein, it binds to the Gq protein-coupled TRH receptor and is able to induce MEKK activation via the PKC pathway, as well as Ras-dependent MEKK activation via tyrosine phosphorylation of the Shc protein. This in turn induces ERK activation via phosphorylation of MEK and increases the expression of prolactin. In addition, TRH also activates inositol phospholipids and causes an influx of Ca^2+^, leading to an increase in the amount of intracellular Ca^2+^ and activation of Ca^2+^-dependent protein kinases, which accelerate prolactin release [11,12,13].

In addition to TRH, there are many transcription factors and protein complexes that regulate prolactin synthesis at the transcription level; PIT-1 is a pituitary-specific transcription factor, and as the most essential transcription factor for prolactin expression, it positively regulates the transcription of prolactin and growth hormone [14]. Epidermal growth factor is able to stimulate prolactin expression in and release from lactating cells through estrogen receptor-α [15]. The transcriptional regulator FOXO1 controls the expression of the growth hormone prolactin and is a key regulator of pituitary function [16]. In addition, estradiol is involved in the TRH-induced prolactin secretion through the PI3K/Akt pathway [17]. Olfactory marker protein (OMP) can influence prolactin synthesis and secretion by controlling Ca^2+^ and TRH signaling [18]. CEBPD is able to bind to the promoter of prolactin to inhibit prolactin secretion and cell proliferation [19]. The data in all of these studies suggest that transcriptional regulation of prolactin is mediated by many factors; however, there is a lack of systematic studies on the regulatory role that TRH has on the pituitary transcriptome.

Small noncoding RNAs known as microRNAs (miRNAs) are involved in the post-transcriptional control of gene expression [20]. By attaching to the 3′UTR segment of the target gene mRNA, miRNAs suppress mRNA production or cause it to be degraded [21]. MiRNAs have been linked to a number of biological processes, including cell division and proliferation [22], apoptosis [23], tumor suppression [24], and immunological control [25]. In recent years, an increasing number of studies have shown that miRNAs play a particularly important role in regulating the synthesis and secretion of reproductive hormones. For example, norepinephrine can influence the synthesis of FSH and LH through miR-7 [26], miR-361-3p can target FSHβ to regulate FSH [27], Gh1 mRNA expression can be suppressed by miR-543-5p, which also lowers GH secretion [28], and miR-200s regulate LH expression by targeting wt1a [29]. In addition, miR-9 promotes prolactin production by targeting dopamine receptors [30], miR-130a inhibits PRL expression by targeting estrogen receptor α [31], and miR-375 promotes PRL synthesis through Rasd1 and Esr1 [32]. These data imply that miRNAs are essential regulators of PRL production and secretion. There is, however, limited research on the role that miRNAs play in the control of TRH-mediated PRL production and secretion. The GH3 cell line was isolated and established from rat adenotrophic tumors [33] and is often used as a growth hormone-producing cell model as well as a prolactin-producing cell model because of these cells ability to secrete growth hormone and prolactin [11]. In this study, we examined the mechanism by which TRH controls PRL production and secretion using GH3 cells as a model.

In this work, we treated male rats with TRH and found that TRH was able to regulate PRL synthesis and secretion. We performed in vitro RNA-seq experiments using the GH3 cell line as a model to analyze the differentially expressed RNAs before and after TRH treatment and to screen for genes that regulate prolactin synthesis and secretion at the transcription level. Then, we used software to predict miRNAs targeting the 3′UTR of *PRL* mRNA and investigated which miRNAs could regulate prolactin synthesis and secretion post-transcription by dual luciferase reporter, cell transfection, RT-qPCR, and Western blotting experiments. Our study revealed multiple regulatory pathways that may be involved in the regulation of PRL production and secretion by TRH and we further provide evidence that miRNAs directly regulate PRL synthesis.

## 2. Results

### 2.1. TRH Treatment Can Promote the Synthesis and Secretion of PRL in Rats

To investigate the effect of TRH on the synthesis and secretion of PRL, we first treated rats with TRH. The adenohypophysis and blood samples were collected. The results demonstrated that after TRH therapy, PRL mRNA expression was considerably increased in the pituitary tissue of rats (Figure 1A), and the secretion of prolactin increased significantly in rat plasma (Figure 1B). We then treated rat pituitary cells with TRH, and the results showed a substantial increase in *PRL* mRNA expression (Figure 1C), a significant increase in PRL protein expression (Figure 1D), and a significant increase in prolactin secretion (Figure 1E). Next, we performed the same treatment on the GH3 cell line, and the experimental results were the same as those obtained after treating rat pituitary cells with TRH (Figure 1F–H).

### 2.2. Analysis of Differentially Expressed Genes before and after TRH Treatment

To investigate the changes in mRNA in rat adenohypophysis before and after TRH treatment, we performed RNA-Seq. After sequencing was completed, we first assessed the quality of the raw data (Appendix A), and in addition, to ensure the data quality, the raw data were processed using Trimmomatic to generate clean data (Appendix A). Then, the sequences were compared with the reference genome, and the comparison results were collected (Appendix A).

To compare the differentially expressed mRNAs before and after TRH treatment, we used *p* value < 0.05 and a difference of |log_2_fold change| > 0.5 as the criteria for mRNAs with significant differential expression (Figure 2A and Appendix A). We identified 590 differentially expressed mRNAs, of which 102 mRNAs were significantly upregulated and 488 mRNAs were significantly downregulated (Figure 2B and Appendix A). We used GO and KEGG analyses to examine the enrichment of all mRNAs and performed functional analysis of the differentially expressed mRNAs. The results of GO enrichment analysis are shown in Appendix A. KEGG analysis showed that the upregulated pathways were the MAPK signaling pathway, the C-type lectin receptor signaling pathway, the cGMP-PKG signaling pathway, and the NF-kappa B signaling pathway (Figure 2C and Appendix A). The downregulated pathways included the retrograde endocannabinoid signaling pathway, the cGMP-PKG signaling pathway, the glucagon signaling pathway, and the MAPK signaling pathway (Figure 2D and Appendix A). Among these, the MAPK and cGMP-PKG signaling pathways are common signaling pathways. In addition, we verified some of the genes that are regulated by TRH, and we identified genes that could be involved in the regulation of prolactin secretion by RT-qPCR. The RT-qPCR results were consistent with the expression trends seen in the RNA-seq data (Figure 2E).

### 2.3. miR-126a-5p Targets the 3′UTR of PRL mRNA

To investigate whether non-coding RNAs can be involved in the TRH-mediated regulation of PRL synthesis and secretion, we first used software to predict miRNAs that might target the 3′UTR of rat *PRL* mRNA, and then we identified five candidate miRNAs, namely, miR-105, miR-126a-5p, miR-300-5p, miR-382-5p, and miR-764-3p (Appendix A). These miRNAs all have the potential to target the 3′UTR of rat *PRL* mRNA. We next transfected the constructed pmiR-PRL-3′UTR-WT reporter plasmid with the predicted miRNA mimics and found that the transfection of miR-126a-5p mimics resulted in a more than 30% reduction in luciferase activity (Figure 3A). Then, based on the base complementation information predicted from TargetScan (Figure 3B), we mutated the target complementation sequence to construct a mutant reporter gene plasmid, pmiR- PRL-3′UTR-MUT. After transfection, the relative luciferase activity of the mutant reporter vector was significantly restored compared with that of the wild-type reporter vector (Figure 3C). In addition, we found that the expression of miR-126a-5p was significantly downregulated in rat adenohypophysis tissue, rat adenohypophysis cells, and GH3 cells after TRH treatment (Figure 3D).

### 2.4. miR-126a-5p Regulates the Level of PRL mRNA Expression and PRL Secretion in GH3 Cells

To further verify the regulation of prolactin by miR-126a-5p, we performed cell transfection. Apoptosis was detected by flow cytometry, and the results showed that the effect of the transfection reagent on cells was negligible (Figure 4A). To demonstrate the transfection effect, we examined the expression level of miR-126a-5p in GH3 cells using RT-qPCR, and the results showed that mimic transfection resulted in a significant increase in the level of miR-126a-5p expression and a significant decrease in the level of miR-126a-5p expression after transfection with inhibitor (Figure 4B).

As a positive control, we transfected PRL siRNA into GH3 cells, and as expected, the results showed that the expression level of *PRL* mRNA was significantly decreased after transfection with PRL siRNA (Figure 4C). We then transfected miR-126a-5p mimics and inhibitor into GH3 cells. RT-qPCR assays showed that miR-126a-5p overexpression significantly downregulated *PRL* mRNA expression, and inhibition of miR-126a-5p expression resulted in significant upregulation of *PRL* mRNA expression (Figure 4D). Meanwhile, Western blotting results showed that miR-126a-5p overexpression led to a significant decrease in the level of PRL protein expression, and inhibition of miR-126a-5p expression led to a significant increase in the level of PRL protein expression (Figure 4E). In addition, ELISA results showed that miR-126a-5p overexpression led to a significant decrease in PRL secretion, and inhibition of miR-126a-5p expression led to a significant increase in PRL secretion (Figure 4F). Furthermore, RT-qPCR assays showed that TRH treatment upregulated *PRL* mRNA expression; however, overexpression of miR-126a-5p can reduce the increase of *PRL* mRNA expression by TRH treatment (Figure 4G). The trend of ELISA results was consistent with RT-qPCR results (Figure 4H). In addition, we examined the expression of ZBTB20, Esr1, and Pit1, which are key transcription factors for prolactin. Transfection with the miR-126a-5p mimic had no significant effect on the mRNA expression levels of ZBTB20, Esr1, or Pit1 (Figure 4I). These results suggest that miR-126a-5p can inhibit *PRL* mRNA expression and reduce PRL synthesis and secretion, and TRH can regulate PRL by regulating miR-126a-5p in GH3 cells.

### 2.5. miR-126a-5p Regulates the Level of PRL mRNA Expression and PRL Secretion in Rat Pituitary Cells

To ensure the accuracy of the transfection results, the same experiments were performed in rat pituitary cells to verify the regulation of prolactin by miR-126a-5p. The results of flow cytometry showed no significant difference in apoptosis between the control and transfected groups (Figure 5A). RT-qPCR showed that transfection with the mimic resulted in a significant increase in miR-126a-5p expression, and transfection with the inhibitor resulted in a significant decrease in miR-126a-5p expression in rat pituitary cells (Figure 5B).

We transfected PRL siRNA into rat pituitary cells as a positive control, and RT-qPCR results showed that the expression level of *PRL* mRNA was significantly decreased after transfection with PRL siRNA (Figure 5C). Then, we transfected miR-126a-5p mimics, an inhibitor, into rat adenopituitary cells. RT-qPCR results showed that miR-126a-5p overexpression resulted in a significant downregulation of *PRL* mRNA expression, and inhibition of miR-126a-5p expression resulted in a significant upregulation of *PRL* mRNA expression (Figure 5D). Western blotting results showed that miR-126a-5p overexpression resulted in a significant decrease in the level of PRL protein expression, and inhibition of miR-126a-5p expression resulted in a significant increase in the level of PRL protein expression (Figure 5E); the ELISA results showed that miR-126a-5p overexpression led to a significant decrease in PRL secretion. Furthermore, inhibition of miR-126a-5p expression led to a significant increase in PRL secretion (Figure 5F). In addition, RT-qPCR assays showed that TRH treatment increased *PRL* mRNA expression, however, overexpression of miR-126a-5p can reduce the increase of PRL mRNA expression by TRH treatment (Figure 5G). The trend of ELISA results were consistent with RT-qPCR results (Figure 5H). These results suggest that miR-126a-5p can inhibit *PRL* mRNA expression and reduce PRL synthesis and secretion and TRH can regulate PRL by regulating miR-126a-5p in rat pituitary cells.

## 3. Discussion

TRH was originally named for its ability to promote the production of thyrotropin from rat pituitary cells [34], and upon further study, it was discovered that TRH causes rapid prolactin release from the anterior pituitary cells of rats [35]. TRH is used as a common prolactin-releasing factor that regulates prolactin synthesis and secretion while participating in the regulation of mammalian reproduction. Additionally, TRH treatment of GH3 cells increases prolactin secretion from these cells and TRH also plays a crucial role in prolactin-producing cell models [11]. To investigate the role that TRH mRNA expression played in the regulation of PRL secretion in rats, we examined the expression of PRL in rats after TRH tail vein injection, examined the changes in prolactin expression in GH3 and rat adenohypophysis cells after TRH treatment, and examined the mRNA expression in GH3 cells before and after TRH treatment using RNA-seq. We identified 590 differentially expressed genes, and a related study showed that TRH treatment significantly upregulated PRL gene expression in GH3 cells [36]. Our sequencing results also showed that after TRH therapy, PRL expression levels in GH3 cells were noticeably increased. Therefore, we further analyzed the changes in the mRNA expression of GH3 cells after TRH treatment to provide a basis for future studies. In addition, in our previous study, we found that differential miRNA expression before and after sexual maturation could regulate pituitary hormone secretion, so we also investigated the effect of post-transcriptional regulation on prolactin synthesis and secretion.

For the differentially expressed mRNAs, RNA-seq results showed that genes such as those encoding Nppc, Fgf1, PRL, Cd63, Npw, and Il23a were significantly upregulated, while genes such as those encoding Lats1, Cacna2d1, Top2a, and Tfap2a were significantly downregulated in GH3 cells after TRH treatment. Some of these differentially expressed mRNAs that are regulated by TRH have been verified in other studies; for example, a previous study showed that Nppc expression was significantly increased in GH3 cells after treatment with 100 nM TRH or 10 µM trichothecene [37]. Other studies have shown that upregulation of specific genes, as determined by sequencing results, affect prolactin synthesis and secretion. For example, FGF is a physiological regulator of prolactin production and secretion [38]. Npw is a neuropeptide with neuroendocrine functions, and IVC infusion of Npw promotes prolactin release in rats [39,40]. Npw expression significantly increased after TRH treatment, so we speculate that Npw is involved in the TRH-induced PRL release. Lats1, a tumor suppressor homolog, affects the promoter activity of cellular Gh1 and Prl in pituitary tumors, and Lats1 is involved in the regulation of prolactin expression [41]. Cacna2d1 is a key gene that regulates voltage-gated Ca^2+^ channels and contributes to hormone secretion in pituitary cells [42,43]. These genes are involved in the synthesis and secretion of prolactin through different pathways.

In addition, we performed enrichment analysis of differentially expressed mRNAs in GH3 cells after TRH treatment, and among the upregulated genes, GO enrichment showed that the biological process category, “ERK1 and ERK2 cascade” (GO:0070371), was the most enriched process. TRH binds to the G-coupled TRH receptor and activates the ERK signaling pathway through a cascade reaction, increasing prolactin gene expression. Among the molecular function categories, “hormone activity” (GO:0005179) was the most significantly enriched function, and hormone activity is inextricably linked to the synthesis and secretion of prolactin. GO analysis suggests that differentially expressed RNAs may be involved in the process of pituitary biological function through different processes that regulate the secretion of pituitary hormones.

KEGG enrichment data showed that the significantly upregulated genes were mainly involved in the MAPK signaling pathway, C-type lectin receptor signaling pathway, cGMP-PKG signaling pathway, arachidonic acid metabolism, and glycerophospholipid metabolism. It has been shown that pregnenolone sulfate can upregulate PRL production and secretion in vivo and in vitro through MAPK signaling [44]. cGMP levels increase with increased prolactin release [45], and the cGMP-PKG signaling pathway is involved in the regulation of the intracellular Ca^2+^ concentration in GH3 cells, which in turn affects prolactin secretion [46]. In addition, exogenous arachidonic acid at 10 nM to 1 μM stimulates PRL secretion in GH3 cells [47], and phospholipid metabolism is involved in prolactin hormone secretion following TRH treatment [48]. Negative crosstalk exists between NF-kappaB and the prolactin receptor/JAK2/STAT5 activation pathway [49]. A comprehensive review has been published which explains the regulation of TRH-induced prolactin release and synthesis by different signaling cascades, and it covers the function of TRH in prolactin synthesis in GH3 cells [11]. Significantly downregulated genes were mainly enriched in signaling pathways such as the cell cycle and oxidative phosphorylation, as well as pathways involved in Parkinson’s disease, amyotrophic lateral sclerosis, and Alzheimer’s disease. Some of these downregulated signaling pathways are associated with inhibition of prolactin secretion. It has been reported that TRH can regulate the activity of dopaminergic neurons through neurosynapses and subsequently regulate prolactin secretion [50]. Intracerebroventricular angiotensin administration decreases the level of prolactin in the plasma [51]. In addition, dopamine is able to inhibit prolactin secretion through the pertussis toxin signaling pathway [52]. Dopamine, a common neurotransmitter that causes Parkinson’s disease, can bind to type-2 dopamine receptors to inhibit prolactin secretion [8]. KEGG pathway analysis showed that multiple signaling pathways work together to promote prolactin synthesis and secretion within GH3 cells after TRH treatment.

In addition, miRNAs have been linked to a growing range of physiological processes in animals, including cell differentiation, immunological control, organogenesis, and cancer. The most significant endocrine organ in living things is the pituitary gland, and studies have indicated that miRNAs are involved in pituitary hormone release. In a prior investigation, we discovered that GnRH controls FSH production and secretion via miR-488 [53] and that miR-186-5p can target the 3′UTR of FSHb to regulate FSH secretion from the rat adenohypophysis [54]. Although miR-9, miR-130a, and miR-375 have been shown to regulate the synthesis and secretion of prolactin, none of these miRNAs directly target PRL. In this study, we report for the first time that miR-126a-5p can target the *PRL* mRNA 3′UTR. This was determined by targeting prediction software as well as a dual luciferase reporter system, which demonstrated that miR-126a-5p is a regulator of PRL gene expression. In addition, TRH treatment decreased the expression of miR-126a-5p in rat pituitary tissue, GH3 cells, and adenopituitary cells, and TRH was able to promote not only PRL gene expression but also PRL synthesis and secretion, so we hypothesized that TRH could regulate *PRL* mRNA expression by regulating miR-126a-5p.

In recent studies, it has been demonstrated that miR-126a-5p contributes to a number of important pathological disorders. In mice, miR-126a-5p has been demonstrated to inhibit the production of ADAMTS-4 and prevent the development of abdominal aortic aneurysms; during angiotensin (Ang) II-induced AAA formation, Ang II leads to downregulation of miR-126a-5p, which is able to directly target ADAMTS-4, leading to elevated ADAMTS-4 expression, which in turn causes AAA formation in mice. miR-126a-5p-ADAMTS-4 is a potential therapeutic target for abdominal aortic aneurysm development [55]. Furthermore, it has been demonstrated that miR-126a-5p is linked to the emergence of certain disorders in humans. For instance, miR-126a-5p plays a role in the chronic pulmonary hypertension that is caused by hypoxia in newborns [56], and miR-126a-5p plays an essential role in protecting the integrity of the blood-brain barrier; microglial miR-126a-5p is able to attenuate blood-brain barrier disruption by targeting the inhibition of matrix metallopeptidase 9. Auranofin ensures the integrity of the blood-brain barrier by increasing the expression of miR-126-5p while improving experimental autoimmune encephalomyelitis [57]. With continuous research on miRNAs, miR-126a-5p has gradually been shown to serve as a diagnostic marker for diseases as well as a drug target. Stroke and hypertension can both be treated by targeting miR-126a-5p. Potential pharmacological targets for the treatment of liver failure include miR-126a [58], and miR-126a-5p has promising potential to be developed as a new drug because it downregulates DLK1 and promotes CD4 T-cell differentiation toward Th1. A very promising biomarker for the CE diagnosis of cystic echinococcosis may be hsa-miR-126a-5p [59]. miR-126a-5p can be used to assess pancreatic injury in the acute phase of pancreatitis [60], and it may be involved in the development of congenital obstructive nephropathy [61]. miR-126a-5p is capable of directly targeting NOX2 to prevent brain injury and has neuroprotective effects against ischemic stroke [62]. These studies suggest that miR-126a-5p affects cell proliferation and differentiation and can have a protective effect on tissues and organs while playing a crucial role in disease detection and prevention. Since the GH3 cell line is derived from pituitary adenoma cells, we hypothesized that miR-126a-5p could be used as a therapeutic target for pituitary adenoma. However, it is not clear whether miR-126a-5p plays a role in hormone secretion, so our study further explored the potential functions of miR-126a-5p and verified the role of miR-126a-5p in the regulation of prolactin secretion by TRH in GH3 cells. miRNA biosynthesis and degradation are regulated by various factors, such as DNA. In addition, noncoding RNAs such as circRNAs and lncRNAs can also bind competitively with miRNAs to regulate gene expression [63], and this will be the focus of our future research considering that the mechanism of miRNA regulation by TRH is too complex.

In summary, our study revealed that genes such as those that encode Nppc, Fgf1, Cd63, and Npw play important roles in TRH-mediated prolactin regulation, and genes such as those that encode Lats1, Cacna2d1, Top2a, and Tfap2a can be used as therapeutic targets for disease. It was also demonstrated that miR-126a-5p can target the *PRL* mRNA 3′UTR to inhibit *PRL* mRNA expression as well as prolactin secretion. In contrast, exogenous TRH treatment was able to downregulate miR-126a-5p while upregulating PRL expression. Based on this, we propose that TRH can regulate the synthesis and secretion of PRL through miR-126a-5p. We finally mapped the mechanism of TRH regulation of PRL synthesis and secretion in the rat adenohypophysis (Figure 6). Our study further refines the molecular mechanism of TRH regulation of PRL secretion and provides support for the role of miRNA in regulating hormone secretion.

## 4. Materials and Methods

### 4.1. Animals and Ethics

This study was approved by the College of Animal Science of Jilin University and the Institutional Committee for Animal Protection and Utilization of Jilin University (license number: SY202110121). Male Sprague-Dawley rats that were 8 weeks old and in good health were purchased from Liaoning Changsheng Biotechnology Co., Ltd (Liaoning Province, China). and kept in a secure animal facility (temperature 22–25 °C, 12 h light/12 h dark environment). The rats were given standard chow and free access to drinking water. All experimental procedures were performed in strict accordance with animal welfare ethics and animal welfare laws and regulations.

### 4.2. Cell Culture

The GH3 rat pituitary tumor cell line was purchased from the Peking Union Cell Resource Center. GH3 cells were cultured in F10 medium containing 15% horse serum (HS), 2.5% fetal bovine serum (FBS), and 1% penicillin and streptomycin (Gibco). The GH3 cell line was cultured in an incubator at 37 °C with 5% CO_2_. The GH3 cell line was passaged three to four times per week at a multiplicity of 1:2 to 1:4. GH3 cells that were utilized for additional experiments were harvested between the 6th and 8th generation.

Rat adenopituitary cells were isolated and cultured from the pituitary tissue of 8-week-old male SD rats. Dulbecco’s modified Eagle’s medium/F12 (DMEM/F12, HyClone, Logan, UT, USA), which contains 10% fetal bovine serum (FBS, Gibco, New York, NY, USA) and 1% penicillin/streptomycin, was used to cultivate rat adenopituitary cells. The cells were cultured at 37 °C in a 5% CO_2_ incubator. In a previous investigation, our team provided a detailed description of the procedure for isolating and cultivating rat adenohypophysis cells [64]. Every experiment was carried out in an aseptic environment, and we moved the cells from the growth flasks to 12-well plates before transfection.

### 4.3. Rat TRH Treatment

One milligram of TRH (TRH, Selleck) was mixed well in 10 mL of saline to create a storage solution at a 100 μg/mL concentration. The storage solution was diluted into 5 μg/mL injection solution and set aside. Twelve male SD rats aged 8 weeks were split into control and experimental groups, each with six rats; 200 μL of saline was injected intravenously into the control group, and 200 μL of the 5 μg/mL injection solution was injected intravenously into the experimental group [65]. Ten minutes later, a CO_2_ anesthetic machine was used to euthanize the rats, and both blood and pituitary tissues were obtained.

### 4.4. TRH Treatment of Cells

Trypsin-EDTA solution A was used to digest the cells. GH3 cells were centrifuged at 1000× rpm for 5 min (adenopituitary cells were centrifuged at 200× *g* for 10 min) to collect the cells, and they were subsequently resuspended in F10 containing 15% HS and 2.5% FBS (adenopituitary cells were treated with DMEM/F12 containing 10% FBS). Cells were added at a density of 1.0 × 10^6^ cells per well into 12-well plates. After 24 h of culture in a 37 °C incubator with 5% CO_2_, the cells were treated with TRH. To create a medium containing 100 nM TRH, TRH was dissolved in serum-free baseline medium. The supernatant was collected after cells were treated with medium containing 100 nM TRH, and RNA was subsequently extracted.

### 4.5. RNA-Seq

After treating GH3 cells with TRH, we collected the cells from the control (N1, N2, N3) and treated groups (T1, T2, T3) for RNA-Seq. To generate total RNA samples for subsequent library construction, we performed cell fragmentation, nucleic acid extraction and purification, and quality control on the samples from GH3 cells. The RNA concentration was measured using a Qubit 2.0 instrument (Invitrogen, Q32866), and RNA integrity and genomic contamination were analyzed on agarose gels. After the samples met the quality requirements for library sequencing, the Illumina Xten platform was used for sequencing.

### 4.6. Data Evaluation and Quality Control

Information such as the quality value of raw data was collected, and the quality of the sequencing data was visually assessed using FastQC 0.11.2. To ensure the quality of the information analysis, quality clipping was performed by Trimmomatic to ensure accurate and valid data.

### 4.7. RNA-Seq Evaluation

The data collected from the samples were first compared to the reference genome using the mapping information from HISAT2 2.1.0. Then, RSeQC 2.6.1 was used for the redundant sequence analysis and the insertion fragment distribution analysis based on the matching results. Next, Qualimap 2.2.1 was used to check the homogeneity distribution and to analyze the genomic structure distribution based on the matching results. The distribution of sequences on the chromosomes was determined using BEDTools 2.26.0, as well as by statistical analysis of gene coverage.

### 4.8. Gene Structure Analysis

BCFtools 1.5 was used to identify possible SNP loci based on mapping results, and SnpEff 2.36 was used to determine the effect of SNP loci on genes. Sequences mapped to the genome were assembled using StringTie 1.3.3b and then compared with known gene models using GffCompare 0.10.1 to identify novel transcribed regions. Variable shear analysis and fusion gene analysis were performed using ASProfier 1.0.4 and EricScript0.55, respectively.

### 4.9. Expression Level Analysis

We used StringTie software and known gene models to assess the expression levels of target genes. To make the estimated gene expression levels comparable across genes and experiments, we used TPM (transcripts per million) to assess the gene expression level, as TPM takes into account the effects of sequencing depth, gene length, and sample quality on read counts.

### 4.10. Analysis of Differential Expression

We used DESeq 1.26.0 for the analysis of differentially expressed genes. To determine the genes that had a significantly different expression, we set the screening conditions as *p* value < 0.05 and difference multiplicity |log_2_fold change| > 0.5. The results of this differential analysis were visualized using volcano plots and clustering heatmaps, and cluster analysis was performed.

### 4.11. Gene Enrichment Analysis

To identify the set of differentially expressed genes, we performed gene enrichment analysis. KEGG pathway and KOG classification enrichment analyses were performed using clusterProfiler. The findings of the gene function enrichment analysis were used for association analysis and to further create network maps.

### 4.12. Prediction of Potential PRL Gene Targets

We used TargetScan, miRanda, and RNAhybrid native software to predict miRNAs that may target the 3′UTR of *PRL* mRNA. miRNAs with duplicate target sites were discarded, and the predictions of the three software programs were merged.

### 4.13. Cell Transfection

Cells were inoculated into 12-well plates at a density of 3 × 10^5^ cells per well and transfected using the Ribo FECTTMCP Transfection Kit (Guangzhou, China) according to the manufacturer’s protocol. miRNA mimics, inhibitors, and NC were transfected at a final concentration of 100 nm in a 37 °C incubator containing 5% CO_2_. The siRNA, mimics, and inhibitors were obtained from RiboBio (Guangzhou, China), (mimic negative control:F:5′-UUUGUACUACACAAAAGUACUG-3′, R:3′-AAACAUGAUGUGUUUUCAUGAC-5′;inhibitor negative control:5′-mCmAmGmUmAmCmUmUmUmUmGmUmGmUmAmGmUmAmCmAmAmA-3′,m refers to 2′-Ome;miR-126a-5p mimic:F:5′-CAUUAUUACUUUUGGUACGCG-3′, R:3′-GUAAUAAUGAAAACCAUGCGC-5′;miR-126a-5p inhibitor:5′-mCmGmCmGmUmAmCmCmAmAmAmAmGmUmAmAmUmAmAmUmG-3′,m refers to 2′-Ome) and the transfection process, and treatment methods were performed strictly according to the manufacturer’s recommended protocol.

### 4.14. RNA Extraction and Real-Time Quantitative PCR

After cell treatment, we extracted RNA from GH3 and rat pituitary cells using TRIzol reagent (Tiangen, Beijing, China). We then measured the purity and concentration of the extracted RNA using a NanoDrop ND-2000 spectrophotometer (NanoDrop Technologies, Beijing, China) to ensure RNA quality. We then used MonScriptTM RTIII All-in-One Mix with dsDNase (Mona Bio, Wuhan, China) and a FastQuant RT Kit (with gDNase, Tiangen, China) for reverse transcription to obtain cDNA. Next, MonAmpTM ChemoHS qPCR Mix (Mona Bio) was used, and real-time q-PCR was run to detect the mRNA expression of relevant genes. All primers used in PCR and real-time q-PCR are shown in Appendix A. GHPDH was used as an internal reference control gene for mRNA, and U6 was used as an internal reference control gene for miRNA.

### 4.15. Dual Luciferase Reporter Gene Assay

The pmiR-PRL-3′UTR-WT plasmid was constructed with the assistance of RiboBio (RiboBio, Guangzhou, China) and confirmed by sequencing. The targeting of miRNA to the 3′UTR of *PRL* mRNA was verified using the pmiR-RB-REPORT™ dual luciferase reporter vector. The reporter fluorophore of this vector is the sea kidney luciferase gene (hRluc), and the firefly luciferase gene (hluc) is used as an internal reference to correct the fluorescence. The 3′UTR of *PRL* is cloned downstream of the *hRluc* gene, and the miRNA mimics are transformed at the same time with the constructed vector in cells. A decrease in the relative fluorescence value of the reporter gene indicates the interaction of miRNAs with target genes.

### 4.16. Protein Blotting Analysis

After transfection, we lysed the cells using RIPA lysis buffer (Yase, Shanghai, China) containing 1 mM phenylmethylsulfonyl fluoride (PMSF). We determined the concentration of protein samples using the BCA Protein Assay Kit (Beyoncé, Shanghai, China) according to the manufacturer’s instructions. After protein separation by PAGE gel electrophoresis, proteins were transferred to polyvinylidene difluoride (PVDF) membranes (Millipore, Carlsbad, CA, USA). Then, PVDF membranes were closed at room temperature for 15 min with rapid closure solution (Yase), and the PVDF membranes were incubated in diluted specific primary antibody at 4 °C overnight. The next day, PVDF membranes were washed 3 times (10 min each time) in TBST buffer and then incubated in secondary antibody at room temperature for 1 h. After washing, the membranes were treated with an Omni-ECL™ Ultra-Sensitive Chemiluminescence Detection Kit (Ya Enzyme, Shanghai, China) and developed on a Tanon 5200 Multi System (Tanon, Shanghai, China). The protein blotting bands were analyzed in grayscale using ImageJ 1.52a. The antibodies used in this experiment were anti-GAPDH (1:1000; #2118, Cell Signaling Technology, Shanghai, China, 2118S), anti-PRL (1:1000, Abcam, Cambridge, MA, USA, ab183967), and anti-rabbit IgG (H+L)-HRP (1: 3000; Cell Signaling Technology, Shanghai, China, 7074S).

### 4.17. Enzyme-Linked Immunosorbent Assay

The collected blood samples and cell supernatant were centrifuged at 1000× *g* for 20 min, and the supernatant was collected. Changes in PRL levels were detected using a rat PRL ELISA kit (EnzymeLink Biotechnology Co., Ltd., Shanghai, China). The kit was removed from the refrigerator 60 min in advance and equilibrated to room temperature (18–25 °C). Standard wells, blank wells and sample wells were set up with 50 μL of different concentrations of standards and added to the standard blanks, 50 μL of the diluted sample were added to the blank wells and 50 μL of the test sample were added to the sample wells. Then, 100 μL of horseradish peroxidase (HRP)-labeled detection antibody was added to each well, and the reaction wells were sealed and incubated for 60 min at 37 °C. After incubation, the liquid was discarded, and 350 μL of washing solution was added to each well five times. Then, 50 μL of each substrate solution (A and B) was added and incubated for 15 min at 37 °C, and the reaction was terminated by adding 50 μL of the termination solution. The optical density (OD) of each well at 450 nm was measured immediately using a Teccan Infinite M200 Pro Multimode plate reader. The concentration of the standards was used as the horizontal coordinate, and the corresponding OD value was used as the vertical coordinate to plot the linear regression of the standards and to calculate the concentration of each sample.

### 4.18. Apoptosis Analysis

The apoptosis of GH3 and rat pituitary cells was detected by flow cytometry. After transfection for 24 h and 48 h, the adnexal cells were digested with trypsin EDTA solution, and the GH3 cells were centrifuged at 1000× rpm for 5 min to collect cells, while pituitary primary cells were centrifuged at 200× *g* for 10 min to collect cells. The collected cells were resuspended in 500 μL of 1× working solution (5× binding buffer diluted with double-distilled water into 1× working solution). Five microliters of membrane-linked protein V-fluorescein isothiocyanate (FITC), and 10 μL of propidium iodide (PI) were added to a blank control tube and a sample tube containing the above cell suspension, with the sample tube containing the membrane-linked protein V-FITC/PI apoptosis kit (Multi Sciences, Hangzhou, China). Apoptosis was detected by flow cytometry (BEKCMAN COULTER) for 1 h.

### 4.19. Statistical Analysis

All data are the mean ± standard deviation of three independent biological replicates. The *t* test of GraphPad Prism 9 was used to compare the significance of the two groups of data, and the multiple comparison data were analyzed for significance using SPSS 19.0 one-way ANOVA, where *p* < 0.05 was considered statistically significant.

## Figures and Tables

**Figure 1 ijms-23-15914-f001:**
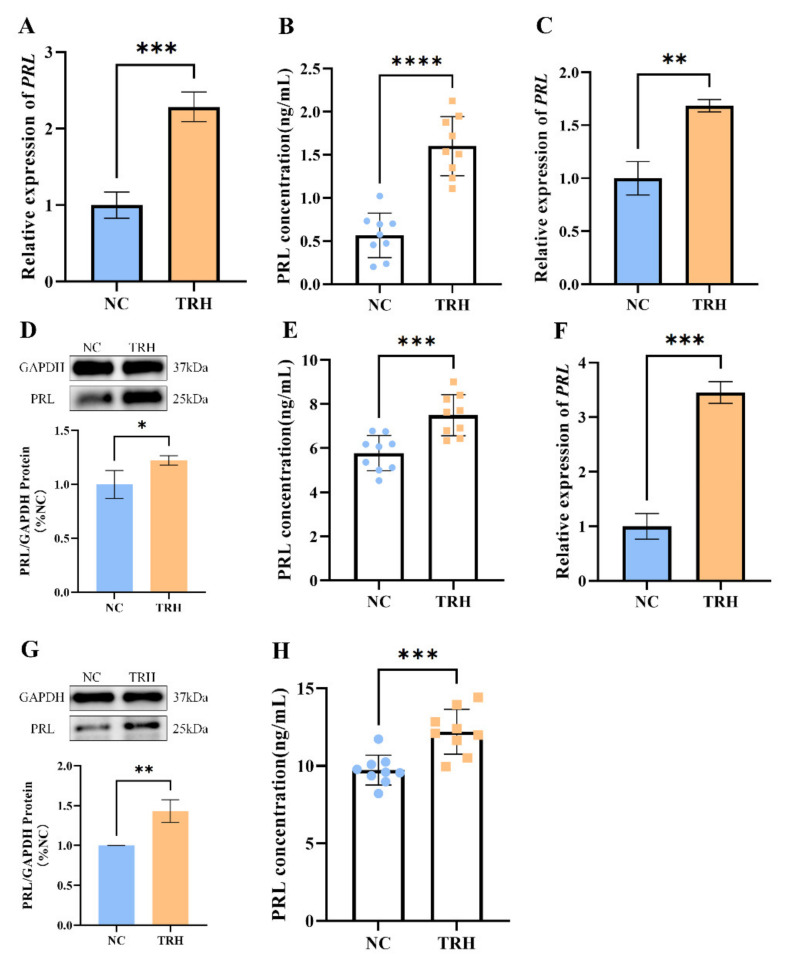
TRH treatment promotes the synthesis and secretion of PRL in rats. (**A**) Changes in *PRL* mRNA expression in rats after TRH treatment. (**B**) Changes in PRL secretion in rats after TRH treatment. (**C**) Changes in *PRL* mRNA expression in rat adenopituitary cells after TRH treatment. (**D**) Changes in PRL protein expression in rat adenopituitary cells after TRH treatment. (**E**) Changes in the level of PRL secretion in rat adenopituitary cells after TRH treatment. (**F**) Changes in the level of PRL secretion in rat adenopituitary cells after TRH treatment. (**G**) Changes in *PRL* mRNA expression in GH3 cells after TRH treatment. (**H**) Changes in the level of PRL secretion in GH3 cells after TRH treatment. *, *p* < 0.05; **, *p* < 0.01; ***, *p* < 0.001; ****, *p* < 0.0001.

**Figure 2 ijms-23-15914-f002:**
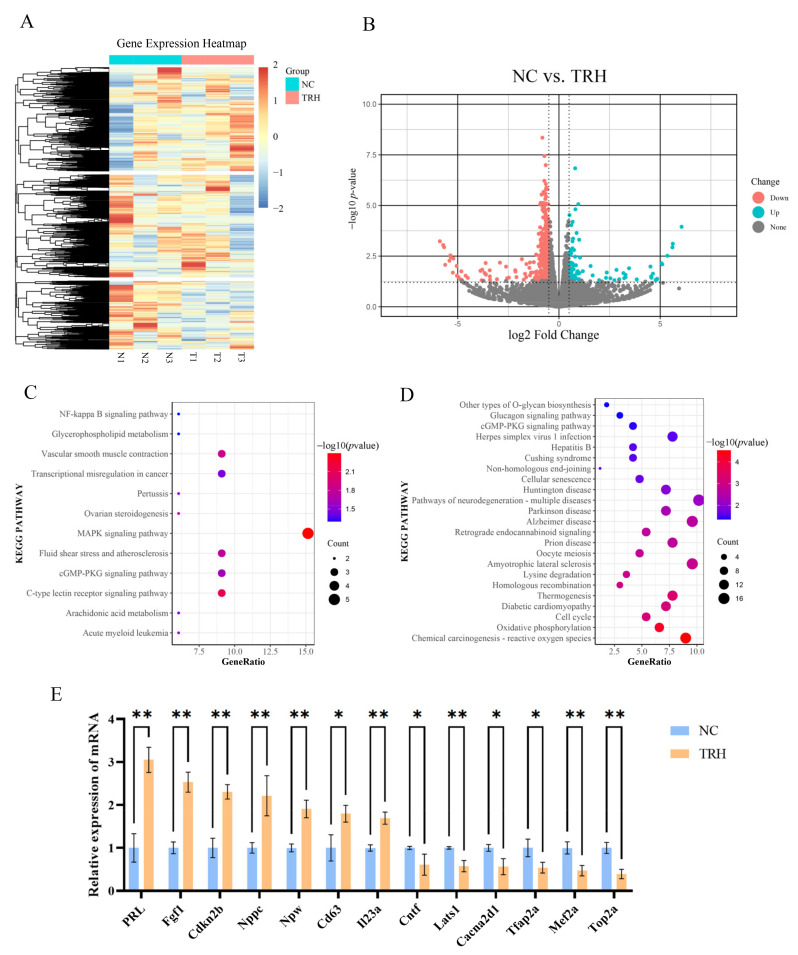
Analysis of differentially expressed genes before and after TRH treatment. (**A**) Cluster analysis of mRNAs that exhibit differential expression. Different rows indicate various mRNAs, while different columns represent various samples. The mRNA expression levels of the samples are shown as different colors (log2TPM+1). (**B**) A volcano plot comparing the levels of mRNA in the two groups. The logarithm of the difference in mRNA expression between the two groups is represented by the horizontal coordinate. The negative logarithm of the *p* value is represented by the vertical coordinate. Red dots indicate mRNAs that have been downregulated, green dots indicate mRNAs that have been upregulated, and gray dots indicate mRNAs that have had little to no change in expression. (**C**,**D**) Analysis of differentially expressed genes using KEGG pathway enrichment. The horizontal coordinate displays the percentage of mRNAs ascribed to this pathway in relation to the total number of annotated genes, and the vertical coordinate indicates the KEGG metabolic pathway. (**E**) Genes with differential expression as determined by RT-qPCR. *, *p* < 0.05; **, *p* < 0.01.

**Figure 3 ijms-23-15914-f003:**
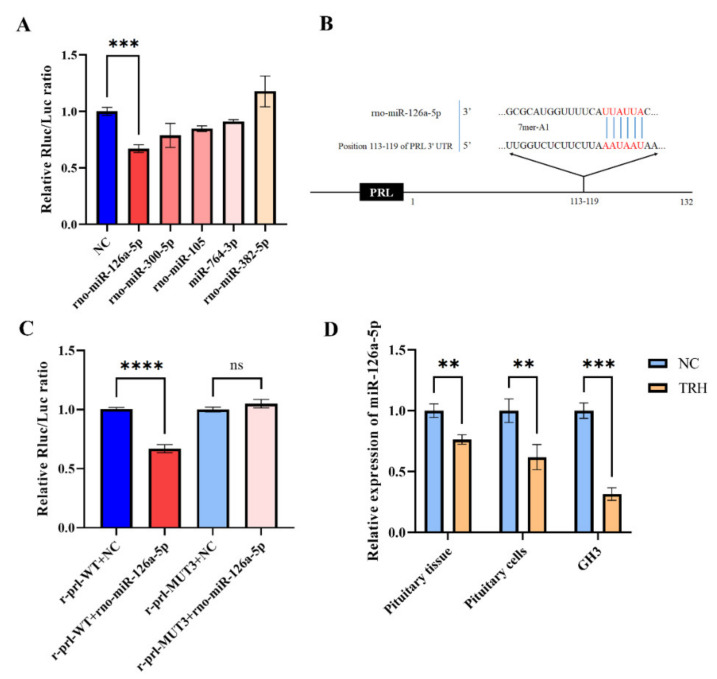
miR-126a-5p targets the 3′UTR of *PRL* mRNA. (**A**) Identifying miRNAs that might directly target the 3′UTR of *PRL* mRNA. The five potential miRNAs were transfected into the pmiR-PRL-3′UTR-WT vector, and relative luciferase activity was measured. (**B**) TargetScan’s anticipated miR-126a-5p base complementary pairing sequence with the 3′UTR of *PRL* mRNA is highlighted in red. (**C**) Changes in relative luciferase activity after the transfection of plasmids with NC/mimics. (**D**) Changes in miR-126a-5p expression after TRH treatment of rats, rat adenopituitary cells, and GH3 cells. ns = no significant, *p* > 0.05; **, *p* < 0.01; ***, *p* < 0.001; ****, *p* < 0.0001.

**Figure 4 ijms-23-15914-f004:**
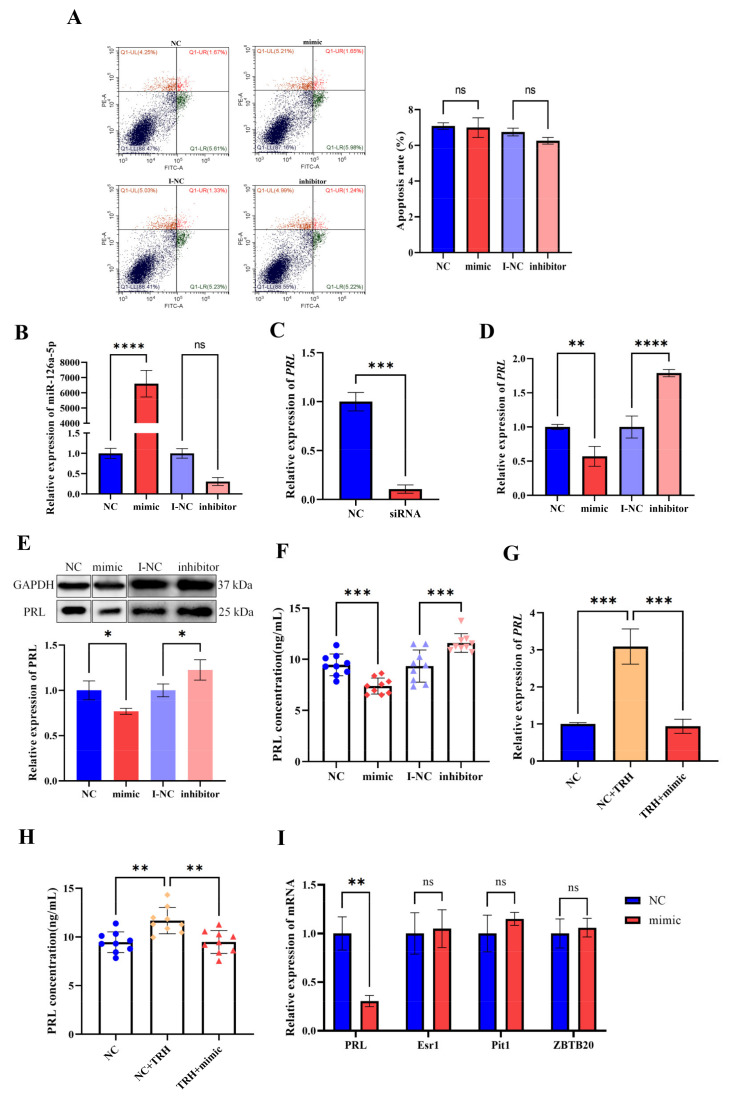
miR-126a-5p regulates the level of *PRL* mRNA expression and PRL secretion in GH3 cells. (**A**) Apoptosis rate in GH3 cells after transfection. (**B**) Relative expression of miR-126a-5p in GH3 cells after transfection with a miR-126a-5p mimic/inhibitor. (**C**) Relative expression of *PRL* mRNA in GH3 cells after transfection with PRL siRNA. (**D**) Relative expression of *PRL* mRNA in GH3 cells after transfection with a miR-126a-5p mimic/inhibitor. (**E**) Changes in PRL protein expression in GH3 cells after transfection with a miR-126a-5p mimic/inhibitor. (**F**) Changes in PRL secretion in GH3 cells after transfection with a miR-126a-5p mimic/inhibitor. (**G**) Relative expression of *PRL* mRNA in GH3 cells after TRH treatment and transfection with a miR-126a-5p mimic. (**H**) Changes in PRL secretion in GH3 cells after TRH treatment and transfection with a miR-126a-5p mimic. (**I**) Changes in the mRNA expression of PRL transcription-related genes. ns = no significant, *p* > 0.05; *, *p* < 0.05; **, *p* < 0.01; ***, *p* < 0.001; ****, *p* < 0.0001.

**Figure 5 ijms-23-15914-f005:**
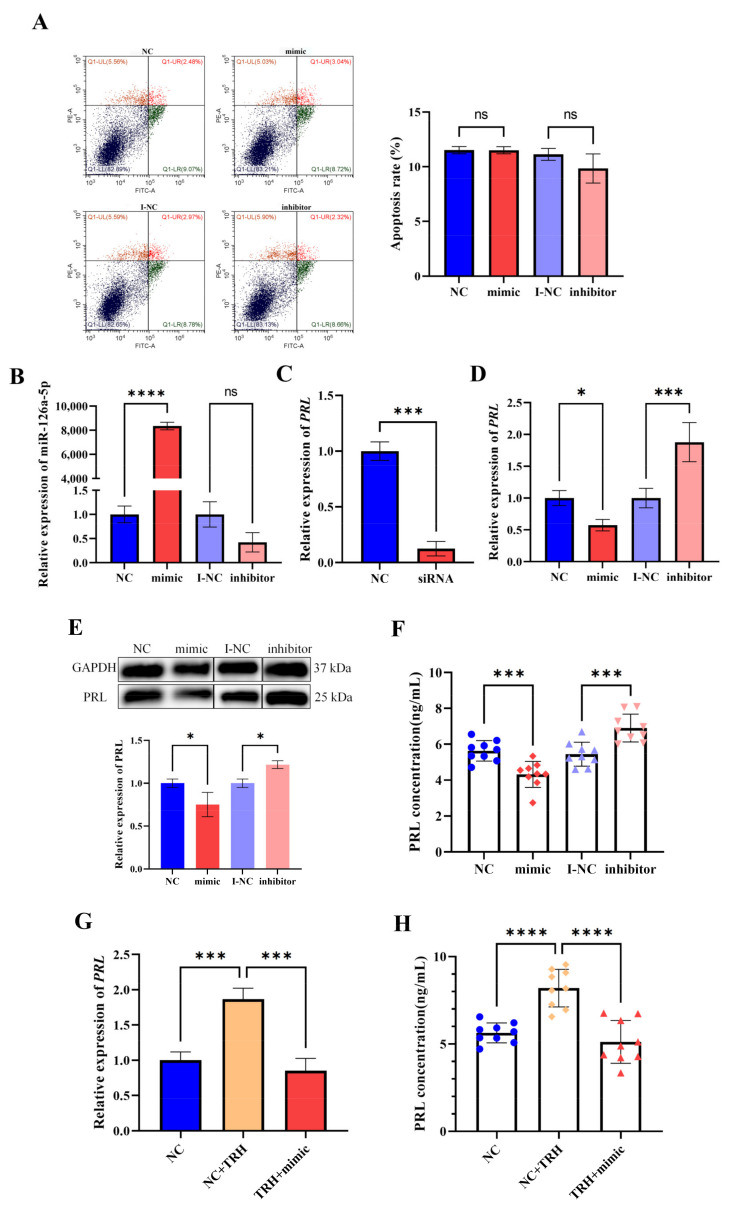
miR-126a-5p regulates the level of *PRL* mRNA expression and PRL secretion in rat adenopituitary cells. (**A**) Apoptosis rate in adenopituitary cells after transfection. (**B**) Relative expression of miR-126a-5p in adenopituitary cells after transfection with a miR-126a-5p mimic/inhibitor. (**C**) Relative expression of *PRL* mRNA in adenopituitary cells after transfection with PRL siRNA. (**D**) Relative expression of *PRL* mRNA in adenopituitary cells after transfection with a miR-126a-5p mimic/inhibitor. (**E**) Changes in PRL protein expression in adenopituitary cells after transfection with a miR-126a-5p mimic/inhibitor. (**F**) Changes in the level of PRL secretion in adenopituitary cells after transfection with a miR-126a-5p mimic/inhibitor. (**G**) Relative expression of *PRL* mRNA in rat adenopituitary cells after TRH treatment and transfection with a miR-126a-5p mimic. (**H**) Changes in PRL secretion in rat adenopituitary cells after TRH treatment and transfection with a miR-126a-5p mimic. ns = no significant, *p* > 0.05; *, *p* < 0.05; ***, *p* < 0.001; ****, *p* < 0.0001.

**Figure 6 ijms-23-15914-f006:**
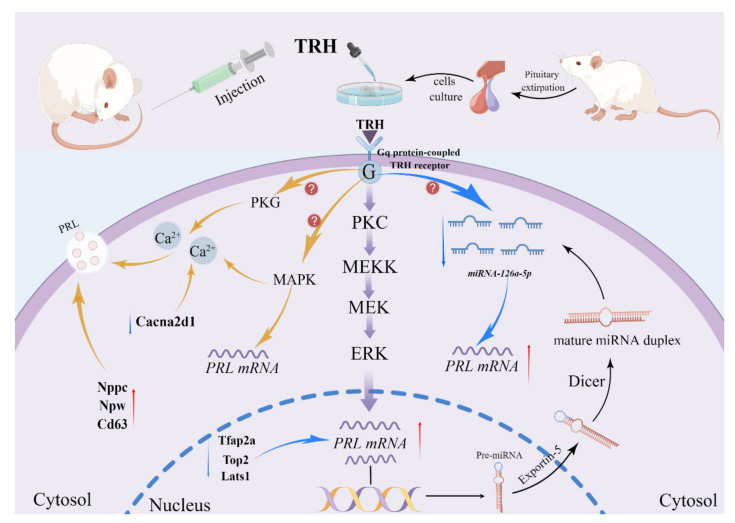
Mechanism of TRH regulation of PRL synthesis and secretion in the rat adenohypophysis. The blue arrows in the diagram represent inhibition, the yellow arrows represent facilitation, the purple arrows represent known signaling pathways, and the black arrows represent the brief process of miR-126a-5p synthesis. This figure is drawn using Figdraw (https://www.figdraw.com/static/index.html#/, accessed on 16 November 2022). The unique authorization code is IIUYIe6020.

## Data Availability

Not applicable.

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
