# Peer review of "TRH Regulates the Synthesis and Secretion of Prolactin in Rats with Adenohypophysis through the Differential Expression of miR-126a-5p"

_ijms, 2022, doi:10.3390/ijms232415914_

Round 1

Reviewer 1 Report

This manuscript from Dr. Wen-Zhi Ren team was well presented, and the work seems to be orchestrated logically. Although the manuscript maybe acceptable in its current form through proper justification, it can be improved by addressing the below concerns.

1. With 12 animals used in this study, why only male rats were included. Although the reports of using male rats for these types of studies is not uncommon, inclusion of equal number of males and females could have benefitted this study.

2. Besides 126a-5p, there are four other miRNAs that have the same target. Most of these miRNAs seem to be poorly conserved, including 126a-5p. On these lines, could the effects of one miRNA be precluded by any of the other four in-vivo, especially when it is used as therapeutic?

3. There is a wealth of information and reports that are available now on miRNAs. From a clinical point of view, details about how 126a-5p can be beneficial in both cancerous and non-cancerous conditions could add more value to this manuscript.

Reviewer 2 Report

In this manuscript, Zhao et al. performed RNA-seq to measure the impact on the transcriptome of GH3 cells following treatment with the thyrotropin-releasing hormone (TRH). While it was previously known that TRH treatment leads to an increase in prolactin (PRL) synthesis and secretion, the current work is the 1st one to report global changes in the transcriptome. Besides the PRL gene, the authors found that many other genes involved in the synthesis and secretion of PRL are also affected. Further, the authors decides to focus on miRNAs which have been reported to also regulate PRL in some way and identified miR-126a-5p as the first direct binder of the 3’ UPR region of the PRL gene which was confirmed via a dual luciferase reporter assay. Through follow-up experiments, the authors showed that TRH treatment leads to a decrease in the expression of miR-126a-5p and an increase in PRL. Overall I recommend the publication of this manuscript in Int. J. Mol. Sci. once all of the following comments are addressed.

·        The text in Figures 2C and D is unreadable (GO Term analysis). The font size should be increased. The authors could make horizontal graphs instead of vertical ones. Also, the gained information from these 2 panels is minimal and the authors don’t really describe them so they should either be removed or properly explained in both the result and the discussion.

·        What is the inhibitor used in figures 4 and 5? No information is provided besides that it needs to be transfected. It is crucial that the authors indicate what this inhibitor is in the main text and also in the material and methods.

·        In the discussion, the authors talk about the role and function of the differently regulated genes by TRH. However, the discussion about some of the genes does not add any value and makes this reviewer wonder why these genes were selected and claimed to be involved with PRL. For example Top2a, the authors wrote: “Type IIA topoisomerase (Top2a), a nuclear protein, has been a key target for anticancer therapy in recent years due to its involvement in DNA replication and cell division [46]”, while this is true, the connection to PRL is not apparent at all. The same can be said about TFAP2A and CD63.

·        On line 275 of page 12, rats is misspelled as rates.

·        The biggest concern with this work is that the 1st part, where the authors performed RNA-seq, and the 2nd part where miRNA studies are performed, feel as 2 stories with little to no logic to have them together. The authors do not provide a rationale as to why the focus is shifted from change in gene expression to miRNA regulation of PRL only. Are the dysregulated genes transcriptionally controlled by miR-126a-5p? The authors should explain why they choose to focus on miRNA following their obtained RNA-seq data.
